# Cavity Nucleation and Growth in Nickel-Based Alloys during Creep

**DOI:** 10.3390/ma15041495

**Published:** 2022-02-17

**Authors:** Felix Meixner, Mohammad Reza Ahmadi, Christof Sommitsch

**Affiliations:** Institute of Materials Science, Joining and Forming, Graz University of Technology, Kopernikusgasse 24/1, 8010 Graz, Austria; mohammad.ahmadi@tugraz.at (M.R.A.); christof.sommitsch@tugraz.at (C.S.)

**Keywords:** creep cavitation, pore formation, pore growth, classical nucleation theory

## Abstract

The number of fossil fueled power plants in electricity generation is still rising, making improvements to their efficiency essential. The development of new materials to withstand the higher service temperatures and pressures of newer, more efficient power plants is greatly aided by physics-based models, which can simulate the microstructural processes leading to their eventual failure. In this work, such a model is developed from classical nucleation theory and diffusion driven growth from vacancy condensation. This model predicts the shape and distribution of cavities which nucleate almost exclusively at grain boundaries during high temperature creep. Cavity radii, number density and phase fraction are validated quantitively against specimens of nickel-based alloys (617 and 625) tested at 700 °C and stresses between 160 and 185 MPa. The model’s results agree well with the experimental results. However, they fail to represent the complex interlinking of cavities which occurs in tertiary creep.

## 1. Introduction

Nickel-based superalloys are the current paragons of creep resistant metals. They are known for their great strength and toughness plus excellent creep and corrosion resistance. They have been used predominantly in turbine engines during the past century [1]. Newly developed, advanced supercritical thermal power generation facilities [2] are in need of materials to cope with their high operating temperatures of up to 700 °C and pressures up to 350 Bar. Here, nickel-based superalloys are being employed as tubing, heat exchangers and fasteners due to their exceptional microstructural stability at high temperatures. These applications demand long component lifetimes and high safety factors, but the demands made on them for component thickness and weight are fortunately less stringent than those for aviation.

These creep conditions, characterized by low stresses at moderately high homologous temperatures and long creep times lead to diffusion creep [3], which is dominated by the diffusion of vacancies and most commonly leads to intergranular fracture [4,5]. In some cases, single crystals [6] are produced to mitigate this issue, although this is not feasible for large components.

A physically based model explaining how diffusion leads to intergranular failure would be a great help in predicting the lifetimes of materials under these conditions and ensuring safety in their future use. Intergranular fracture has long been explained by the nucleation of cavities on the grain boundary [7], which then coalesce to form large cracks [8]. The exact mechanisms of cavity nucleation, however, are still not clear [9]. Needham et al. [10] found that the nucleation rate of cavities is proportional to the strain rate, a relation which is accepted to this day [11]. Grain boundary sliding, most recently developed by He and Sandström [12,13,14], demonstrates this expected relation.

We propose a model based on classical nucleation theory [15], which was first used by Raj and Ashby [16] and which has been improved with recent new developments by Fernandez-Caballero and Cocks [17] to model the nucleation of nanosized cavities at grain boundaries. The growth of these cavities is then described by diffusion-controlled growth [18], which is similar to equations previously derived by Hull and Rimmer [19]. The simultaneous nucleation of new cavities and growth of existing cavities are simulated in a Kampmann–Wagner framework [20]. Elements of our model have been used in previous research with more [21,22] or less [23] success. An understanding of the material properties affecting the nucleation and growth of pores will help in the development of superior materials. Appropriately defined critical conditions at which tertiary creep, and shortly thereafter, failure, are expected to occur will lead to better predictions of components’ creep lifetimes.

## 2. Materials and Methods

### 2.1. Materials

Two nickel-based alloys, Inconel^®^ Alloy 617 and Inconel^®^ Alloy 625 were chosen for creep testing. Their nominal chemical composition, as defined in Ref [24,25], can be seen in Table 1. The high chromium content provides the excellent oxidation and hot corrosion resistance and the molybdenum in solid solution increases strength. In Alloy 617, additional cobalt also increases strength while the aluminum added combines with nickel to make spherical γ’ precipitates in the matrix. Alloy 625 is designed to have strong and stable microstructure without γ’ precipitates, instead using niobium as a solid solution strengthener [26]. Alloy 625 was solution-treated at 1200 °C for 12 h and quenched in water.

### 2.2. Creep Testing

The creep specimens were machined according to DIN 50,125 Type B 14 × 70 and were loaded at a constant stress and constant temperature of 700 °C until failure according to DIN EN ISO 204. The temperature was selected to represent the service conditions in advanced ultra-super-critical (A-USC) powerplants [2] using nickel-based superalloys and the stress was chosen to induce moderate creep times between 5000 and 20,000 h. Creep conditions and time to rupture of the specimens are shown in Table 2 and Figure 1, and they show ordinary scatter for creep experiments [27] and a non-monotonic relation between stress and creep time.

### 2.3. Density Measurements

Small sections of the specimens’ shafts near the fracture surface were cut off and weighed in ethanol and in air using a Radwag PS210.X2 Precision (Radwag, Radom, Poland) balance and density determination kit (Radwag, Radom, Poland) to determine their density. Three density measurements were averaged per sample.

### 2.4. Secondary Electron Microscopy

To investigate the microstructural damage, the shafts of the specimens were longitudinally cut in half. These specimens were embedded in Struers Polyfast (Struers Inc., Cleveland, OH, USA) and ground and polished, at first with Silicon carbide paper, then with diamond polishing paste (9 μ, 1 μ) and finally with an OPS solution at a low force for several minutes. The specimens were examined by a Zeiss Ultra 55 (Carl Zeiss AG, Oberkochen, Germany) and a Tescan Mira3 FEG (TESCAN ORSAY HOLDING, Brno, Czech Republic) scanning electron microscope (SEM) with excitation voltages of 3 kV and 5 kV, respectively. The operator identified grain boundaries and captured micrographs when cavities were found along them. Using MATLAB’s image processing toolbox and custom functions, the sizes of these cavities were documented and processed. Cavities in secondary electron imaging appeared as dark circles with white annular highlights, caused by the edge effect at the steep cavity edges. Non-spherical cavities were approximated by the curvature of their edges. The average grain size was also determined from these micrographs.

## 3. Model Development

### 3.1. Classical Nucleation Theory

Classical nucleation theory (CNT) is used to calculate the base nucleation rate of cavities at grain boundaries. This theory was developed in the 1920s [28] and formalized by subsequent authors [15,29]. It has shown great promise in modelling precipitate nucleation [30], phase transformations [31] and crystallization [32]. Raj and Ashby [16] were the first to apply CNT to cavity nucleation, inspiring Hirth and Nix [33] and Riedel [34] to further develop it in this context. In CNT, we consider the free energy change, Δ*F*, to nucleate a spherical cluster of particles to be a function of its volume and a driving force, and the interfacial area between the cluster and the matrix and the interfacial energy density. In a first approximation of our implementation, the cluster is a cluster of vacancies, the driving force is the external tensile stress, *σ*, and the interfacial energy density is the free surface energy of the bulk material, *γ*, giving us Equation (1) for the free energy change.
(1)$$\Delta F=-\frac{4}{3}\pi r^{3}\;\sigma +4\pi r^{2}\;\gamma$$

When plotted as a function of the radius, *r*, in Figure 2, a peak is visible at a specific radius that represents a barrier for nucleation. We designate these values as the critical radius, *r**, and the critical free energy change Δ*F**. Clusters smaller than the critical radius will shrink while supercritical clusters will grow as they both minimize their free energy.

Upon sketching this curve and finding the maximum of Equation (1), we arrive at Equations (2) and (3) for the critical radius and critical free energy change, respectively.
(2)$$r^{*}=\frac{2\gamma }{\sigma }$$
(3)$$\Delta F^{*}=\frac{16}{3}\pi \frac{\gamma^{3}}{\sigma^{2}}$$

Subcritical clusters exist in a quasi-constant equilibrium supply, as a result of thermal fluctuations with their prevalence defined by an Arrhenius function of their required free energy [28,29]. The sophistication of CNT is that the nucleation rate of stable clusters is dictated by the number of critical clusters, which become supercritical per unit time. The nucleation rate is given below in Equation (4), where *N_s_* is the number of available nucleation sites, which when multiplied with the exponential term signifies the equilibrium number of critical clusters, *β** is the attachment frequency of vacancies to a critical cluster and *Z* is the so-called Zeldovich factor.
(4)$$I=N_{s}\;exp\left(-\frac{\Delta F^{*}\;}{kT}\right)\;\beta^{*}\;Z\;$$

The vacancy attachment frequency, *β**, as its names implies, expresses the rate at which vacancies present in the bulk attach to the critical cluster and thus, make it supercritical. It is shown in Equation (5) to be linearly dependent on the number of atomic sites at the surface of a critical cluster, represented by its area, *A**, over the average area of an atom, *a*^2^, on the relative prevalence of vacancies in these atomic sites, *X_v_*, and on the jump rate of vacancies between sites, given as the quotient of the self-diffusion coefficient, *D*, and the square of the interatomic distance, *a*.
(5)$$\beta^{*}=\frac{D}{a^{2}}X_{v}\;\frac{A^{*}}{a^{2}}$$

The Zeldovich factor reduces the nucleation rate and accounts for thermal fluctuations at clusters near the critical size, which are more likely to cause them to dissolve rather than to grow. It was derived by its namesake [35] by first considering a steady state distribution of clusters of all sizes with the principles of detailed balance. Assuming then, that supercritical clusters are removed from the system, but still using the terms for dissolution from the steady state led to a differential equation in terms of vacancies, *n*, for the nucleation rate. Inserting the Arrhenius relation and integrating the differential equation near the critical area of maximum free energy with a Taylor expansion up to the second order term, introduced the Zeldovich factor into the equation. More details on its derivation can be found in the original article [35] or in works by Russell [36] and Riedel [34]. *Z* is shown in Equation (6) in its general differential form and simplified with respect to a critical cluster.
(6)$$Z=\sqrt{\frac{-1}{2\pi kT}\;\;{\frac{\partial^{2}\Delta F}{\partial n^{2}}|}_{n^{*}}}=\frac{1}{n^{*}}\sqrt{\frac{\Delta F^{*}}{3\pi kT}}$$

Inserting the terms from Equations (1)–(3), (5), and (6) into Equation (4), we arrive at the final nucleation rate in Equation (7) after simplifying. It is shown that the nucleation rate is proportional to the number of nucleation sites, the diffusion coefficient, and the concentration of vacancies. In most cases the exponential term has the greatest influence on the nucleation rate.
(7)$$I=N_{s}\;exp\left(-\frac{16\pi \gamma^{3}\;}{3\sigma^{2}kT}\right)\;\frac{DX_{v}16\pi \gamma^{2}}{a^{4}\sigma^{2}}\;\sqrt{\frac{a^{6}}{64\pi^{2}kT}\frac{\sigma^{4}}{\gamma^{3}}}=N_{s}\;exp\left(-\frac{16\pi \gamma^{3}\;}{3\sigma^{2}kT}\right)\;\frac{2DX_{v}}{a}\;\sqrt{\frac{\gamma }{kT}}$$

### 3.2. Heterogenous Nucleation at Grain Boundaries

Equations (1)–(7) are valid for the general case of spherical clusters of vacancies in the bulk, referred to as homogenous nucleation and must be adapted to suit nucleation at other sites in the microstructure.

Clusters and cavities formed on grain boundaries are not spherical but lenticular (lens-shaped), as seen in Figure 3. This is due to the pulling force caused by the grain boundary energy density, *γ_gb_*, forming a dihedral angle, *δ*, where the cluster’s surface meets the grain boundary. Balancing the forces of grain boundary energy and free surface energy gives us Equation (8). These clusters are smaller than the spherical ones in the bulk even though they have the same curvature, *r**, at their maximum free energy. The energy barrier is lower [37] due to their smaller surface area and the energy gained by dissolving the prior grain boundary (dashed green line in Figure 2). When nucleation occurs at sites such as grain boundaries, it is called heterogeneous nucleation.
(8)$$\delta =acos\left(\frac{\gamma_{gb}}{2\;\gamma }\right)$$

Diffusion at the grain boundaries, which is several orders of magnitude greater than the diffusion coefficient of the bulk, also promotes nucleation. Grain boundaries also experience a higher stress than the bulk, because their movement is constrained. This increase was calculated by Anderson and Rice [38] for grains containing 6 square and 8 hexagonal faces. Another important consideration is the effect of real defects (dislocations, faults, tilt boundaries) in the microstructure on the free energy. A theory proposed by Fernandez-Caballero and Cocks [17] assumes the total volume of vacancy clusters and defects to be constant, which then allows these defects to supply vacancies to the clusters and increases the final driving force, *σ*, by up to several gigapascals for dislocations intersecting a grain boundary. These modifications are shown in Equation (14). This practically eliminates the typically observed, high critical stress, which is normally needed for nucleation to begin.

We use a theory based on generalized broken bonds [39] to calculate the free surface energy density from the energy of vacancy formation, shown in Equation (15). As a result, it is lower than values from the literature [40], although this is expected for small clusters [41].

### 3.3. Cavity Growth

Hull and Rimmer [19] pioneered the theory of the stress-directed, diffusion-controlled flux of atoms away from cavities, and this has been widely accepted [34,42] even in contemporary literature [13]. This theory also shows surprising similarity to the SFFK model [18], which is based upon Onsager’s principle of maximum entropy production [43]. Equation (9), for the growth rate of a cavity’s radius, *r*, is given below and is also directly proportional to the self-diffusion coefficient, and the number of vacancies in the microstructure. The driving force for growth, *σ*, is reduced by the sintering stress, *2γ/r*, which causes subcritical clusters to shrink.
(9)$$\dot{r}=\frac{D\;X_{v}\;\Omega }{kT\;r}\left(\sigma -\frac{2\gamma }{r}\right)$$

### 3.4. Modelling Implementation

Preliminary calculations showed practically no homogenous nucleation in the bulk, which was expected and observed in the microscopy results. Therefore, we only simulate the nucleation and growth at grain boundaries in detail.

After all corrections the nucleation rate used for the simulations is
(10)$$I=N_{gb}\;exp\left(-\frac{\Delta F_{gb}^{*}\;}{kT}\right)\;\frac{D}{a^{2}}X_{v}\;\frac{A_{gb}^{*}}{a^{2}}\frac{1}{n_{gb}^{*}}\sqrt{\frac{\Delta F_{gb}^{*}}{3\pi kT}}$$
with
(11)$$\Delta F_{gb}^{*}=\frac{\pi {\left(\gamma_{gb}-2\bar{\gamma }\right)}^{2}\left(\gamma_{gb}+4\bar{\gamma }\right)}{3{\bar{\sigma }}^{2}}$$
(12)$$A_{gb}^{*}=\frac{16\pi {\bar{\gamma }}^{2}\left(1-\frac{\gamma_{gb}}{2\bar{\gamma }}\right)}{{\bar{\sigma }}^{2}}$$
(13)$$n_{gb}^{*}=\frac{2\pi \left(\gamma_{gb}{}^{3}-12{\bar{\gamma }}^{2}\gamma_{gb}+16{\bar{\gamma }}^{3}\right)}{3\;a^{3}{\bar{\sigma }}^{3}}$$
and
(14)$$\bar{\sigma }=2.24\;\sigma +\sigma_{D}$$
(15)$$\bar{\gamma }=0.328\frac{Q_{v}}{a^{2}}$$

The model was implemented with MathWorks’ MATLAB in a Kampmann–Wagner framework [20] with the parameters in Table 3. At first, the total grain boundary area density is calculated [44] assuming that all grains are tetrakaidekahedral and of the same size, with the mean grain diameter determined from the SEM images, shown in Table 7.

At every timestep, a new class of cavities is created with a radius equal to *r** from Equation (2). The number density of cavities in this class is the product of the nucleation rate, in Equation (10), and the time interval. All previously existing classes grow their respective cavity radii according to Equation (9). The remaining grain boundary area density, *N_gb_*, that is available for the nucleation of new cavities in the next timestep, is calculated by subtracting the grain boundary area occupied by all existing classes from the total grain boundary area. These steps are repeated until the rupture time of the specimen.

## 4. Results

### 4.1. Density

The densities of all specimens were measured to be lower than that of their corresponding references, as seen in Table 4, with a general trend toward lower densities after longer creep times which is consistent with continuous cavity nucleation and growth. Figure 4 shows a pronounced minimum density in specimens which ruptured after approximately 10,000 h. They may have experienced more creep cavitation due to their combination of high loading and long creep time.

### 4.2. SEM

Over 1100 images were taken from which 250 were analyzed and 814 cavities were measured. Table 5 shows the number of cavities measured in each specimen as well as the size range of cavities measured. Cavity radii were determined to be between a few nanometers, which represents the limit of modern secondary electron microscopy, and a few micrometers. Exemplarily, the simulation results will be validated against the microscopy results from the specimens 31, 33, and 75 considering the large number of cavities found in them to generate statistically valuable results.

In Figure 5 some cavities can be seen along the grain boundaries with their corresponding measured radii overlaid. The external stress is horizontal in all images and most cavities were found at transverse or slightly inclined grain boundaries. Additionally, in most images the larger cavities have begun interconnecting to form microcracks.

### 4.3. Cavity Distributions and Number Densities

Table 6 reveals the phase fraction of cavities in each specimen, which was determined from the density measurements, the mean volume per cavity, *V_mean_*_,_ and the number density of these cavities required to occupy the corresponding phase fraction. The mean volume per cavity is calculated with Equation (16) from the measured cavity radii, *r_i_*, and the number of cavities. Figure 6 shows the distribution of cavity radii and their number density for the selected specimens. The peaks in the histograms shift toward larger radii as creep times increase and cavities have more time to grow. These large cavities may be primarily responsible for the formation of large cracks when they coalesce during tertiary creep.
(16)$$V_{mean}=\frac{1}{n}\overset{n}{\underset{i=1}{\sum }}\frac{4}{3}\pi r_{i}^{3}$$

### 4.4. Simulation Results

Figure 7 shows the simulated results for cavity radii for the selected creep tests. During the first steps of the simulation the nucleation rate is at its nominal maximum. As the grain boundary area is consumed by other existing growing cavities, newer cavities have fewer nucleation sites available, and the nucleation rate decreases. Figure 8 illustrates the growth of the largest cavity, which nucleates at the beginning of creep. As shown in this figure, the size of cavities is almost equal for all specimens, although their initial sizes are not identical and depend on the applied stress. The steady decrease in nucleation rates and the slowing of cavity growth led to an apparent accumulation of large cavities in the histograms. The nominal nucleation rates are listed in Table 7 along with the number of grain boundary nucleation sites available at the beginning of the simulation.

## 5. Discussion

The SEM images show abundant and exclusive grain boundary cavity nucleation as expected, in both the model and the existing literature [48]. Their sizes range from a few nanometers, which is also the limit modern SEM, to a few micrometers. These agree well with results from small angle neutron scattering [49] and microtomography and serial sectioning [50] on creep cavities. Most cavities appear lenticular, as in Figure 3, at all stages of growth until they interconnect with neighboring cavities to form long and thin microcracks. Microcracks were more prevalent in alloy 617 as well as in specimens enduring longer creep times. In order to compare the experimental and simulated results, it is necessary to take into account the error that results from measuring the radii of cavities, which were cut with a plane at an arbitrary height. Figure 9 overlays the cavity size distributions from the experiment (in blue, from Figure 6) with the calculated distributions from Figure 7 when each cavity class is intersected by many planes at uniformly random heights. The simulation agrees quite well with the experiments both in terms of cavity radii and the populations of each size for all but the largest cavities. The largest cavities, which are formed by many cavities combining, cannot be represented in the simulation since it simulates only nucleation and growth, but not with coalescence. To model coalescence would require a deeper physics-based understanding of the interaction of two or more cavities and a suitable framework considering the locations and relative spacing between cavities on a grain boundary surface. This deficiency also explains the discrepancy in the results for the density of the final specimens in Figure 10 where the model predicted lower cavity fractions.

Since the coalescence of cavities is characteristic of tertiary creep, which only represents the short final stage of creep life, it may be adequate to predict creep lifetimes with a “critical” cavity fraction which defines the onset of tertiary creep.

Unfortunately, no detailed strain rate data was recorded for these specimens, making it impossible to contrast these results with other models. Existing empirical or semi-physical models for cavity nucleation predict a linear relation between strain rate and cavity nucleation rate. Grain boundary sliding is one model attributing the nucleation of small cavities to the relative motion of grains [14]. Dislocation pile-ups have also been proposed to nucleate cavities at grain boundaries [9]. While recent advances in calculating creep curves by modelling dislocations are promising [51], dislocation creep plays only a minor role in the long term creep of power plant components. It is more reasonable to model cavity nucleation and growth by diffusion where diffusional creep is the dominant deformation mechanism.

## 6. Conclusions

This study presents a model based on classical nucleation theory, incorporating corrections and improvements, and existing growth models to simulate the complex processes of cavity nucleation and growth in nickel-based alloys during creep. The model was validated by using specimens crept for up to 15,000 h.

The experimental results show cavities at grain boundaries, measured to have radii up to a few micrometers, several of which have interlinked to form cracks in the specimens subjected to longer creep tests. These cavities grow in size and number as creep lifetimes increase.

The model correctly predicts the shapes and relative prevalence of these cavities at the grain boundaries qualitatively, and also the sizes and number densities of the cavities, quantitively. The complex interlinking of cavities in specimens is not simulated and presents the final challenge for this model to predict failure during creep. As such, the final phase fraction of cavities is in better agreement with the specimens exhibiting less cavity interlinking and shorter creep times.

This physics-based modeling will allow the development of new materials with improved creep cavitation resistance and better assessment of creep lifetimes.

## Figures and Tables

**Figure 1 materials-15-01495-f001:**
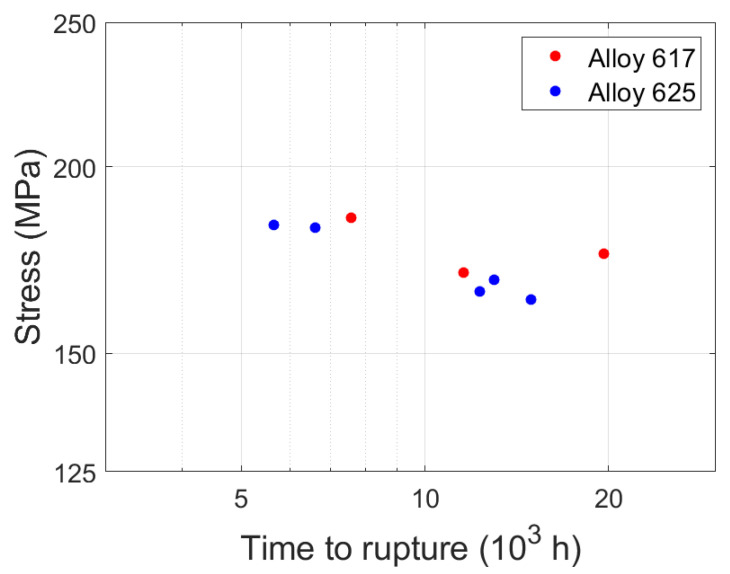
Time to rupture vs. stress for the specimens in this study.

**Figure 2 materials-15-01495-f002:**
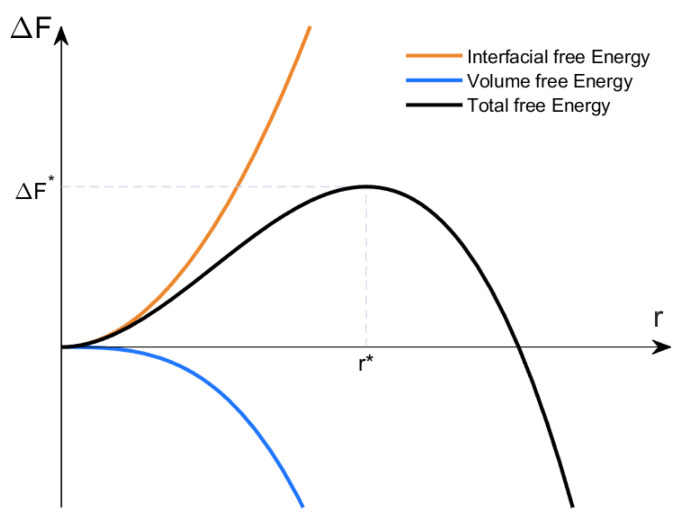
Free energy change vs. cavity radius.

**Figure 3 materials-15-01495-f003:**
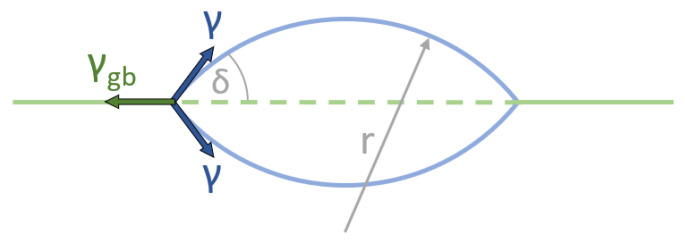
Schematic shape of a cavity at a grain boundary (green line).

**Figure 4 materials-15-01495-f004:**
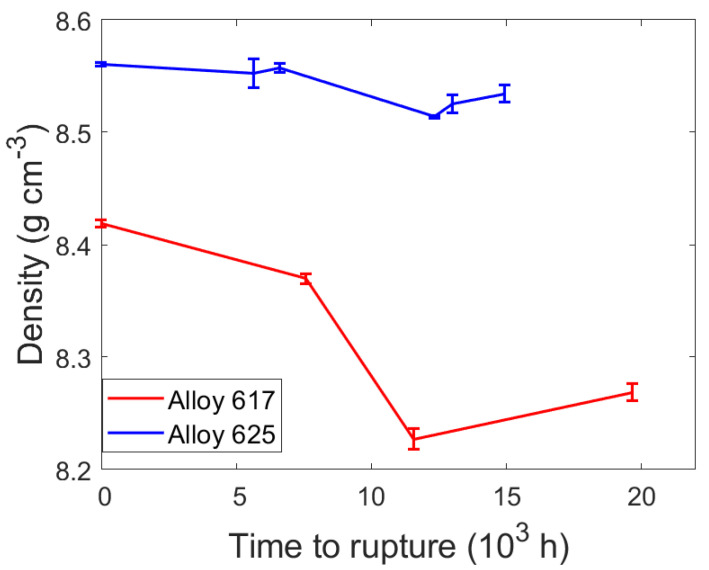
Time to rupture vs. specimen density.

**Figure 5 materials-15-01495-f005:**
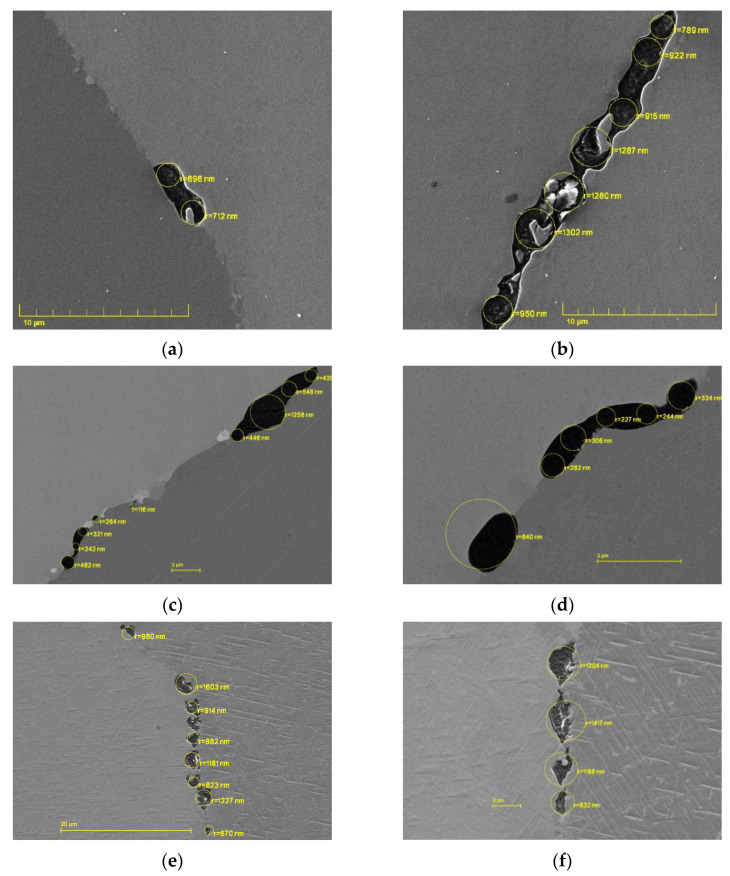
SEM images of cavities at grain boundaries in specimens 31 (**a**,**b**), 33 (**c**,**d**) and 75 (**e**,**f**).

**Figure 6 materials-15-01495-f006:**
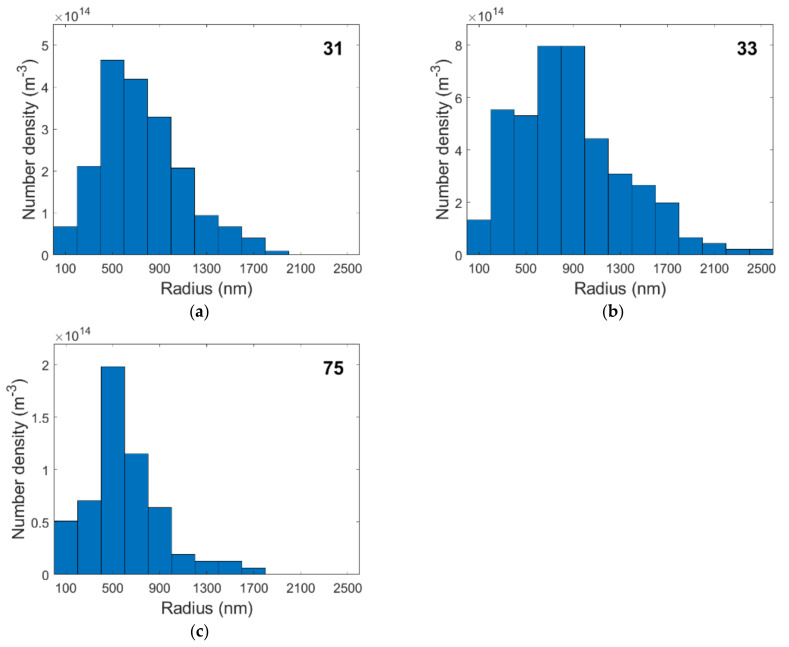
Cavity radii vs. number density in specimens 31 (**a**), 33 (**b**), and 75 (**c**).

**Figure 7 materials-15-01495-f007:**
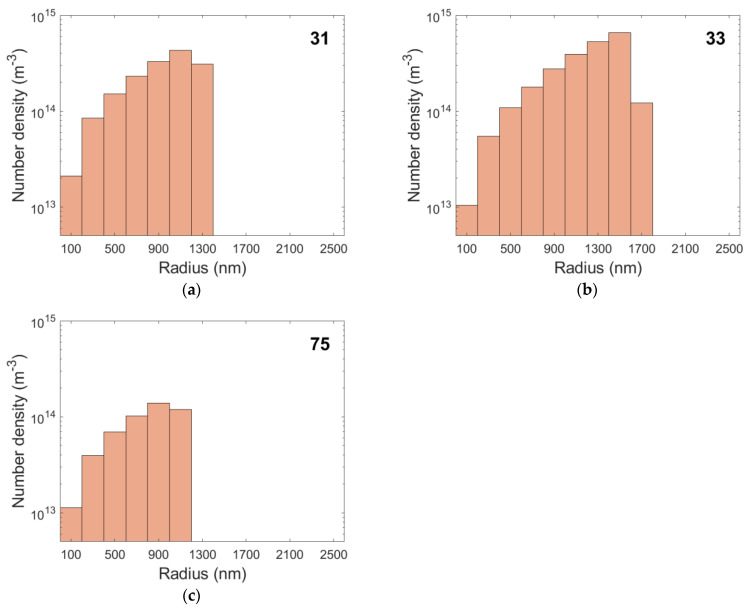
Simulated cavity radii vs. number density for the creep conditions of specimens 31 (**a**), 33 (**b**), and 75 (**c**).

**Figure 8 materials-15-01495-f008:**
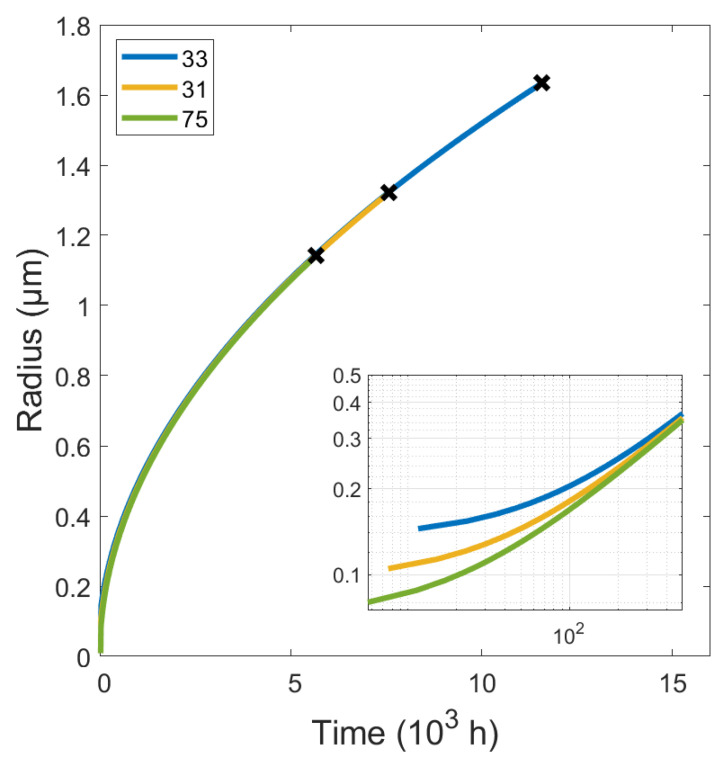
Simulated growth of cavity radii vs. time for the creep conditions of specimens 31, 33 and 75. The x marks the rupture time and end of the simulation. The inset shows the early stage of growth on logarithmic axes.

**Figure 9 materials-15-01495-f009:**
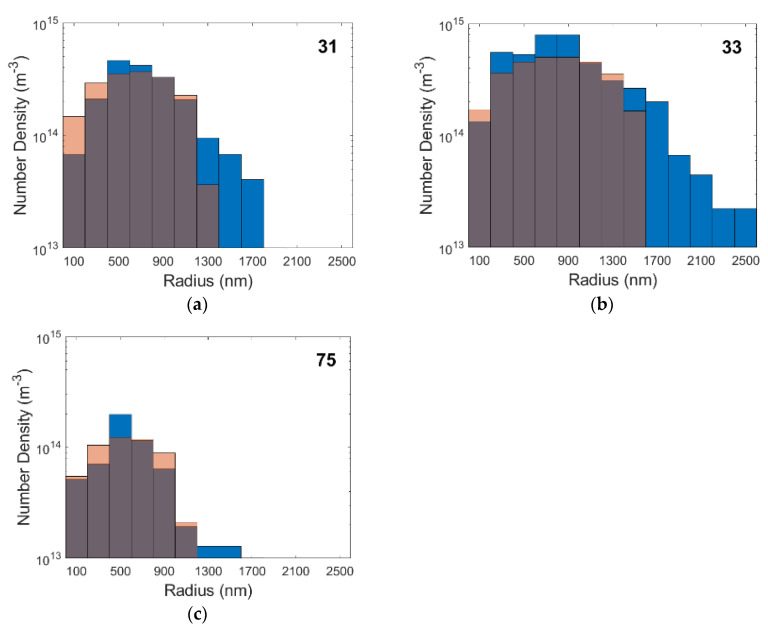
Cavity radii vs. number density of the simulations (red) and experimental (blue) results for specimens 31 (**a**), 33 (**b**), and 75 (**c**).

**Figure 10 materials-15-01495-f010:**
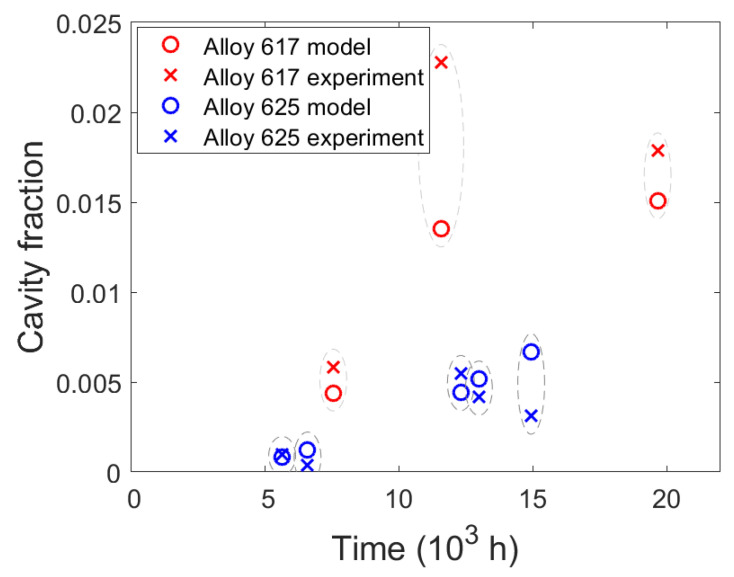
Comparison of cavity fraction vs. time between the simulation (**◯**) and experiments (**X**).

**Table 1 materials-15-01495-t001:** Chemical compositions of the two nickel-based alloys in wt%.

Alloys	Ni	Cr	Mo	Co	Al	Nb	Fe	Mn	Cu	Ti	Si	C	S	P	B
Alloy 617	Bal	21.94	8.64	11.68	1.16		1.02	0.04	0.03	0.39	0.08	0.06	<0.002	<0.002	0.002
Alloy 625	Bal	20.82	8.34	0.005	0.11	3.40	3.29	0.18	0.22	0.12	0.26	0.02	0.014	0.009	

**Table 2 materials-15-01495-t002:** Creep test parameters.

Specimen ID	Material	Temperature (°C)	Stress (MPa)	Time to Rupture (h)	Neck Diameter (mm)
31	Alloy 617	700	185	7558	9.00
32	Alloy 617	700	175	19,656	9.25
33	Alloy 617	700	170	11,577	9.39
71	Alloy 625	700	165	12,317	9.53
72	Alloy 625	700	168	12,990	9.45
73	Alloy 625	700	163	14,940	9.59
74	Alloy 625	700	182	6600	9.07
75	Alloy 625	700	183	5652	9.05

**Table 3 materials-15-01495-t003:** Model parameters and constants.

Parameter	Description	Value
*γ*	Free surface energy of Ni	1.84 J m^−1^ [39]
*σ*	Stress on specimen	Variable (described in Table 2)
*N_s_*	Nucleation sites	Variable (described in Table 7)
*k*	Boltzmann constant	1.380649 × 10^−23^ J K^−1^
*T*	Temperature	973 K (700 °C)
*σ_D_*	Driving force of defects	1.2 × 10^10^ Pa
*γ_gb_*	Grain boundary energy of Ni	0.8 J m^−1^ [45]
*Q_v_*	Vacancy formation energy in Ni	1.7 eV [46]
*D_GB_*	Diffusion coefficient along grain boundaries of Ni	1.511 × 10^−12^ m^2^ s^−1^ [45]
*lp*	Lattice parameter of fcc nickel	0.3499 × 10^−9^ m [47]
*Ω*	Atomic volume	1.0710 × 10^−29^ m
*a*	Interatomic spacing (Ω^1/3^)	0.204 × 10^−9^ m

**Table 4 materials-15-01495-t004:** Specimen densities.

Specimen	Time to Rupture (h)	Density, Mean ± SE (g/cm^3^)	Density Decrease (%)
Alloy 617 reference	-	8.4188 ± 0.003	-
31	7558	8.3698 ± 0.004	0.58
32	19,656	8.2685 ± 0.007	1.79
33	11,577	8.2271 ± 0.009	2.28
Alloy 625 reference	-	8.5607 ± 0.002	-
71	12,317	8.5138 ± 0.001	0.55
72	12,990	8.5251 ± 0.008	0.42
73	14,940	8.5341 ± 0.008	0.31
74	6600	8.5574 ± 0.003	0.04
75	5652	8.5524 ± 0.012	0.10

**Table 5 materials-15-01495-t005:** Measured cavity radii by SEM.

Specimen	Number of Cavities Measured	Radii Range (nm)
31	424	69–1959
32	3	46–121
33	189	101–2942
71	17	17–234
72	18	18–682
73	22	20–113
74	55	3–712
75	86	25–1603

**Table 6 materials-15-01495-t006:** Phase fraction, number density of cavities and mean volume per cavity.

Specimen	Phase Fraction of Cavities (%)	V_mean_ (µm^3^)	Number Density of Cavities (m^−3^)
31	0.58	3.05	1.91 × 10^15^
33	2.27	5.44	4.18 × 10^15^
75	0.10	1.77	5.50 × 10^14^

**Table 7 materials-15-01495-t007:** Nominal nucleation rates and number of nucleation sites.

Specimen	Grain Diameter (m)	N_s_ (m^−3^)	Nucleation Rate (m^−3^s^−1^)
31	200 × 10^−6^	2.65 × 10^23^	6.43 × 10^7^
33	150 × 10^−6^	3.53 × 10^23^	7.58 × 10^7^
75	500 × 10^−6^	1.06 × 10^23^	2.51 × 10^7^

## Data Availability

The data presented in this study are available on request from the corresponding author.

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
