# Peer review of "Cavity Nucleation and Growth in Nickel-Based Alloys during Creep"

_materials, 2022, doi:10.3390/ma15041495_

Round 1

Reviewer 1 Report

  1. There are lots of “Reference source” error in the manuscript.
  2. Why did sample 32 take more time to rupture than that of sample 33? And the same problem for sample 71 and 72.
  3. What is the purpose of section 4.1?
  4. The authors only mentioned the cavities along the boundaries. Are the any intragranular cavities?
  5. More details are needed for the simulation.
  6. Why did the model fail to represent the interlinking of cavities? If so, is it reasonable to use this model to simulate the fracture failure?

Author Response

Thank you for your helpful review. Below are your comments (in Bold) addressed by us.

There are lots of “Reference source” error in the manuscript.

These have been fixed in the document.

Why did sample 32 take more time to rupture than that of sample 33? And the same problem for sample 71 and 72.

This is amount of scatter is normal for creep tests, although it is not often reported in literature.

This could be due to inhomogeneity of the base material from which the specimens are machined, differences in the heat treatment, or in the test apparatus.

The norm for uniaxial creep testing in tension (DIN EN ISO 204) allows up to 38% deviation in times to rupture.

What is the purpose of section 4.1?

Section 4.1 explains the results of the density measurements which are needed to determine the total number of cavities in the specimens

The authors only mentioned the cavities along the boundaries. Are the any intragranular cavities?

No intragranular cavities were observed in the SEM. The model predicts practically no cavities inside the grains and slower growth for them. This is all inline with existing literature on creep cavitation

More details are needed for the simulation.

More details were added to the manuscript in section 3

Why did the model fail to represent the interlinking of cavities? If so, is it reasonable to use this model to simulate the fracture failure?

The model currently only portrays the nucleation and growth of the cavities. The complex processes of interlinking/coalescing cavities when they meet is not currently modeled. This would first need an understanding at the atomic level (molecular dynamics DFT) which could then be integrated into a more sophisticated model which takes into account the positions of cavities on a fictional grain boundary.

Reviewer 2 Report

This manuscript can be recommended for publication in the journal Metals only after major revision taking into account the remarks (see attached file).

Author Response

Thank you for your helpful review. Below are your comments (in Bold) addressed by us.

Lines 8-9 in Abstract: The sentence "Although the share of fossil fueled power plants in electricity generation is declining, their numbers are still rising, making improvements to their efficiency essential" is ambiguous and contradictory. The proposal should be rewritten.

The sentence has been shortened and clarified to:

The number of fossil fueled power plants in electricity generation is still rising, making improvements to their efficiency essential.

Lines 60, 74, 129, 177, 194: There are errors in the display of reference source.

These errors are now corrected.

Line 72: Why creep tests were carried out at the temperature 700 and not under normal conditions? What is the reason for choosing this temperature? The same questions apply to the choice of constant stress value.

This has been clarified in the text:

The temperature was selected to represent the service conditions in advanced ultra super-critical (A-USC) powerplants[21] using nickel-based superalloys and the stress was chosen to induce moderate creep times between 5000 and 20000 hours.

Line 74: Figure 1 should be discussed in more detail.

The scatter and non-monotonic relation has been mentioned in the text:

Creep conditions and time to rupture of the specimens are shown in Table 2 and Figure 1, and they show ordinary scatter for creep experiments[22] and a non-monotonic relation between stress and creep time.

Line 170: From this section, the details of the simulations and the proposed model are not clear. The details of the calculations, the time step and thermodynamic conditions of the simulation as well as the limitations of the proposed model should be discussed in more detail.

More detail and discussion was added to section 3 and section 5.

Line 224 (Fig. 7): How did the simulation achieve times of the order of 10^3 hours and micrometer scales? This question arises because the simulation details are not discussed.

The simulation scales of time and radius are only linked by the eqautions for the nucleation rate and growth rate. Therefore, these scales can have vastly different values.

Chapter 3 on the development of the model now contains a lot more detail.

Authors should choose a common style when denote units of physical quantities. Various units of length (meter, micrometer, nanometer) and time (second, hour) are used.

Hours are the typical units for creep tests in scientific literature. The model uses only SI units internally (seconds, meters) .

Length units in this publication were chosen to be comprehensible. Nanometers and micrometers for microscopy investigations are easily understood. The specimen shaft size in millimeters is a practical unit. The nucleation rate is more abstract and is provided for comparison between the specimens and reproduction of the results.

Line 271: What do the gray histograms in Fig. 8 ahead of the red and blue histograms?

The gray bars represent the overlap of the red and blue histograms. We are open to suggestions on how to better show and compare 2 histograms with one another

Line 280 (Conclusions): The manuscript states that “This study presents a model based on classical nucleation theory…”. However, this model is lost among the results. The authors should discuss and disclose the proposed model in more detail if it is presented for the first time in this work.

The model is now explained better in chapter 3

Reviewer 3 Report

This manuscript presents a model to simulate the cavity nucleation and growth in nickel-based alloys during creep. The experimental and simulation results are compared in order to verify the correctness of the proposed model. The work contains some information of interest. However, the idea of model construction is not very clear, and there are certain errors between the experimental results and the simulation results. As to the language, there are many grammar mistakes throughout the paper, which do not meet the Standard English writing. So, I suggest the rejection of the paper published in Metals.

  1. In lines 37-55, the collation of the existing models and discussion of their  advantages and disadvantages based on literature researches are lack, so that the advantages of the constructed model are unclear.
  2. As seen from Fig. 8 and Fig. 9, certain errors exist between the experimental results and the simulation results. Detailed comments about the errors are necessary, and the limitations and applicability of the model should be given.
  3. In the “Model development” part, compared with the description of the model construction process, it is more important to interpret the purpose and significance of each step of model modification.
  4. In the “Results” and “Discussion” parts, literature comparison and theoretical analysis are scarce.
  5. The language is needed to be improved and refined throughout the paper, which some academic or grammatical errors should be avoided.

Author Response

Thank you for your helpful review. Below are your comments (in Bold) addressed by us.

In lines 37-55, the collation of the existing models and discussion of their advantages and disadvantages based on literature researches are lack, so that the advantages of the constructed model are unclear.

Other theories to model the expected behavior have been mentioned:

However, the exact mechanisms of cavity nucleation are still not clear [9]. Needham et al [10] found that the nucleation rate of cavities is proportional to the strain rate, a relation which is accepted to this day [11]. Grain boundary sliding, most recently developed by He and Sandström [12–14], demonstrates this expected relation.

As seen from Fig. 8 and Fig. 9, certain errors exist between the experimental results and the simulation results. Detailed comments about the errors are necessary, and the limitations and applicability of the model should be given.

The discussion now comments more on this discrepancy

In the “Model development” part, compared with the description of the model construction process, it is more important to interpret the purpose and significance of each step of model modification.

A lot more detail has been added as well as the purpose and effect of the modifications.

In the “Results” and “Discussion” parts, literature comparison and theoretical analysis are scarce.

The discussion comments on the model results and the discrepancy with the investigation. More details have been added.

The language is needed to be improved and refined throughout the paper, which some academic or grammatical errors should be avoided.

The manuscript has been rechecked by two native English speakers working in research science.

It has also been submitted to an English correction service at our university. However, this may take up to 3 weeks.

Reviewer 4 Report

This manuscript presents a model that predicts the shape and distribution of cavities which nucleate at grain boundaries during high temperature creep. Some interesting results are obtained. The manuscript could be of interest to scientific community. 
It is fit for publication with the following minor corrections:

1. In general, the manuscript needs a grammar revision, since several mistakes were observed along it.
2. Tte abstract should be improved.
3. Line 16: Stress and loads are not the same.
2. The authors describe the development of a model for the prediction of creep failures. A novelty of the research work is missing and also a comparison with earlier work is also required.
3. Line 133: why the diffusion coefficient and interatomic distance are taken from literature?
4. Line 165: use the term "equation", not formula.
5. Line 224: Equation 7.
6. In the research they talk about simulation/modeling, but no software or more details are mentioned.
7. According to the developed model, and the experimental results obtained from creep, what is the research perspective?

Author Response

Thank you for your helpful review. Below are your comments (in Bold) addressed by us.

In general, the manuscript needs a grammar revision, since several mistakes were observed along it.

The manuscript has been reviewed by 2 native English speakers working in the natural sciences.

It has also been submitted to an English correction service at our university. However, this may take up to 3 weeks.

The abstract should be improved.

The abstract has been rewritten by a native English speaker

Line 16: Stress and loads are not the same.

Loads was replaced with stresses

The authors describe the development of a model for the prediction of creep failures. A novelty of the research work is missing and also a comparison with earlier work is also required.

The novelty lies in the fact that it is a new physically based model for cavity nucleation. Models of grain boundary sliding and the expected outcome are mentioned in the introduction

Line 133: why the diffusion coefficient and interatomic distance are taken from literature?

It is not feasible to determine these parameters solely for this investigation. This work shows that the model works well with values from literature which allow it to be used with other materials

Line 165: use the term "equation", not formula.

Formula was replaced with equation

Line 224: Equation 7.

The numbering was corrected from 6 to 16 because many equations have now been inserted before it.

In the research they talk about simulation/modeling, but no software or more details are mentioned.

The model is implemented in MATLAB. This has been clarified in the text along with further details about the model

According to the developed model, and the experimental results obtained from creep, what is the research perspective?

The research perspective is to develop new materials which are less susceptible to cavity nucleation and growth and thus, intergranular failure. And to better predict creep life. This has been clarified in the conclusion.

Round 2

Reviewer 1 Report

Is it reasonable to compare the simulation results with the experimental results since the model fail to represent the interlinking of cavities?

Author Response

We believe this is important to show how far the model goes in modelling cavitation. It highlights the final step and an avenue for future research. 
We presume most of the larger cavities and cracks to form only in tertiary creep (the final stage of creep, lasting only about 10% of the total creep time). 
Unfortunately we have no results from interrupted creep tests.

Reviewer 2 Report

In the revised manuscript, the authors took into account all comments. The article can be recommended for publication in present form.

Author Response

Many thanks for your help in the review of our work.